# WorldGPT: Empowering LLM as Multimodal World Model

## ABSTRACT

World models are progressively being employed across diverse fields, extending from basic environment simulation to complex scenario construction. However, existing models are mainly trained on domain-specific states and actions, and confined to single-modality state representations. In this paper, We introduce **WorldGPT**, a generalist world model built upon Multimodal Large Language Model (MLLM). WorldGPT acquires an understanding of world dynamics through analyzing millions of videos across various domains. To further enhance WorldGPT's capability in specialized scenarios and long-term tasks, we have integrated it with a novel cognitive architecture that combines memory offloading, knowledge retrieval, and context reflection. As for evaluation, we build **WorldNet**, a multimodal state transition prediction benchmark encompassing varied real-life scenarios. Conducting evaluations on WorldNet directly demonstrates WorldGPT's capability to accurately model state transition patterns, affirming its effectiveness in understanding and predicting the dynamics of complex scenarios. We further explore WorldGPT's emerging potential in serving as a world simulator, helping multimodal agents generalize to unfamiliar domains through efficiently synthesising multimodal instruction instances which are proved to be as reliable as authentic data for fine-tuning purposes. The project is available on https://anonymous.4open.science/r/WorldGPT-C3B1.

## CCS CONCEPTS

• **Computing methodologies** → **Computer vision tasks**; **Knowledge representation and reasoning**.

## KEYWORDS

Multimodal World Model, Multimodal Large Language Model, Multimodal Data Synthesis

## 1 INTRODUCTION

World models [11, 22] explicitly encapsulate the knowledge of the environment through constructing an internal representation that mirrors external realities. Incorporating with a reliable world model, an agent can discern the laws governing its environment through minimal direct interactions. In the realm of control tasks, which often involve simple and repetitive environments such as virtual simulations [12–14], robotic manipulations [45], and embodied explorations [47], the utility of RNN-based world models has been extensively investigated. These models aid agents in completing

ACM MM, 2024, Melbourne, Australia
© 2024 Copyright held by the owner/author(s). Publication rights licensed to ACM.
ACM ISBN 978-x-xxxx-xxxx-x/YY/MM
https://doi.org/10.1145/nnnnnnn.nnnnnnn

tasks by 'imagining' potential consequences of proposed actions. Given their success in these simplified settings, a critical question arises: can these world models also perform effectively in complex, real-world situations?

Recent advancements in diffusion models [17, 33] have showcased impressive capabilities in generating high-resolution images and videos. These developments have sparked efforts to construct world models applicable to real-world scenarios, particularly in autonomous driving [3, 18, 42]. However, they still fall short of a generalized world model in several key aspects:

- **Limited Scope and Modality Composition**: Training is confined to specific domains and primarily visual states, overlooking the intricate real-world states that encompass multiple modalities.
- **Poor Generalization Ability**: Inference is limited to known scenarios; it cannot reason in unknown situations and struggles with long-sequence problems.
- **Insufficient Dataset Development**: Investigation on how to construct comprehensive, sustainable datasets of world state transitions is still weak, hindering the training and evaluation of world models.

In this paper, we propose **WorldGPT**, a versatile world model capable of freely predicting state transitions across modalities, from any given modality combination to any required modality combination. WorldGPT consists of three components: multimodal encoders that process states from different modalities into unified representations, a Large Language Model (LLM) which predict state transitions in abstract feature space and multimodal decoders that generate the state content in modality space. WorldGPT is trained to harness its inherent textual knowledge and integrate multimodal knowledge through watching millions of internet-sourced videos. To ensure that WorldGPT can understand fine-grained actions and accurately capture the state changes, all videos are preprocessed by the dense video caption model [48] to generate detailed action descriptions with specific time intervals. Then we apply a novel **progressive state transition training** methodology, where the train target is evolved from single modality to multiple modalities, and unimodality to cross-modality. This training approach guarantees the model's effectiveness in complex situations such as missing modalities and combined modalities.

Following the extensive pre-training process, WorldGPT emerges as a holistic world model. However, we have observed that its performance declines in unfamiliar scenarios and tends to forget past information in continuous generation tasks. To mitigate this issue, we drew on theories from cognitive science and design a cognitive architecture [20, 21, 34, 38] for WorldGPT. This framework contains three parts: a knowledge retrieval system which provides external knowledge for special scenarios, an working memory mechanism which manages the history predictions, and a novel **ContextReflector** which efficiently extracts grounded infomation from the retrieved context (i.e., external knowledge and memory). To enable WorldGPT cooperate with the cognitive architecture, we construct

Table 1: A taxonomy of related work on world model.

| World Model | Methods | Modality | Training Resource | Out-of-Domain Prediciton | Continous Prediciton | Output Space |
|---|---|---|---|---|---|---|
| RNN-based | Dreamer[12], DayDreamer[45] | Vision | Simulator | ✗ | ✓ | Embedding |
| Diffusion-based | DriveDreamer[42], UniSim[49] | Vision | Labelled Video | ✗ | ✗ | Pixel |
| Autoregressive-based | GAIA[18], Genie[5] | Vision | Labelled Video | ✗ | ✓ | Both |
| | WorldGPT | Vision, Audio | Unlabelled Video | ✓ | ✓ | Both |

high-quality sequential samples and retrieval-augmented samples to teach WorldGPT to utilize information from retrieved context through the **cognitive-augmented tuning** process. Coupled with the advanced cognitive architecture, WorldGPT's capabilities are further enhanced, allowing it to generalize effortlessly across all tasks.

To foster research in constructing realistic world models, we further present **WorldNet**, a comprehensive dataset for multimodal world state transition. WorldNet comprises two subsets: WorldNet-Wild, constructed through low-cost methods and suitable for pre-training, and WorldNet-Crafted, transformed from high-quality datasets and suitable for thorough evaluation. Specifically, WorldNet-Wild contains millions of samples derived from raw Internet videos, covering a wide range of scenarios and tasks in realist world, varied from outdoor activities like fishing to kitchen chores like kneading the dough. All videos are labelled with machine-generated actions, serving as an extensive training resource for building world models. WorldNet-Crafted is transformed from existing dataset with human-labelled actions. Leveraging the original annotations, we further construct tasks specialized for modality combination prediction, knowledge-enhanced prediction and long-sequence prediction, thereby establishing a holistic evaluation benchmark. We conduct a thorough evaluation of WorldGPT based on WorldNet-Crafted. The results demonstrate WorldGPT's proficiency in modeling world dynamics.

With the capability to process any modality in any domain, WorldGPT can serve as a universal world simulator. Unlike previous generation models that only support simulating static scenes (e.g., Stable Diffusion [33]) or image transformations based on naive editing instructions (e.g., InstructPix2Pix [4]), WorldGPT is capable of synthesizing dynamic scenes that change according to complex interactions, offering more practical significance. Utilizing WorldGPT, we explore a novel learning paradigm for multimodal agents (e.g., MLLMs [52, 58]), namely **dream tuning**, where agents acquire specialized knowledge from WorldGPT to enhance their performance on specific tasks by fine-tuning on synthetic multimodal instruction data. The generation process for this data is efficient: a powerful LLM (e.g., GPT-4) generates the textual components of instructions and drives WorldGPT to complete the multimodal part. We conducted experiments with four widely-used MLLM agents across three diverse tasks ((*Visual Understanding*, *Embodied Planning*, and *Audio-Video Question Answering*). Results reveal that agents trained on synthesized data exhibit competitive performance compared to those trained on authentic data, strongly supporting the reliability of WorldGPT as a world simulator.

Our contributions can be summarized as follows:

- **Development of WorldGPT**: We propose WorldGPT, a generalist world model trained on millions of videos through

a progressive state transition training process, which naturally supports input and output across any combination of modalities.

- **Innovative Cognitive Architecture**: We introduce a novel cognitive architecture tailored for world models, encompassing memory offloading, knowledge retrieval, and ContextReflector.

- **Construction of WorldNet**: We present WorldNet, an comprehensive dataset for world state transitions, ideal for training and evaluating world models.

- **Novel Learning Paradigm for Multimodal Agent** : We explore a new learning paradigm wherein multimodal agents can efficiently acquire knowledge from WorldGPT through dream tuning on synthesized data.

## 2 RELATED WORK

**World Models.** World models have been well adopted in model-based reinforce learning [11–14]. Through predicting future states of the environment based on current and past observations, world models enable agents learn complex behaviors with fewer interactions with the actual environment. With the rapid development of visual encoding and synthesis techniques, from Convolutional Neural Network (CNN) to Diffusion Model (DM), world models' application have evolved from naive simulation environment to complicated real-world scenarios. DriveDreamer and UniSim [42, 49] implicitly learns the visual state transition pattern by learning conditional video generation, where the model is trained to generate future videos (i.e. future states) conditioned on past videos (i.e. past states) and driving actions. GAIA-1, WorldDreamer and Genie [5, 18, 43] considers world modelling as a state-action sequence modeling problem, where they convert all inputs to tokens in an unified space and employ transformer to do next token (i.e. state) prediction in an auto-regressive manner. Table 1 summarizes the similarities and differences of various world models. Models labelled with ✓ in '*Continuous Prediction*' column can receive history information but may not use it properly without specialized training.

**Multimodal Data Synthesis.** While extensive studies focus on generating textual instructions for a broad range of tasks [31, 44, 51], the synthesis of multimodal instruction data is still confined to the following applications: 1) Given real images, generate textual instructions with LLM [27, 29]. For instance, LLaVA [27] and Macaw-LLM [29] pre-collect multimodal data from existing datasets and then prompt ChatGPT [1] to generate textual instructions. 2) Given collected captions, generate images with generation models [8, 36]. The synthesized image-caption pairs can then serve as reliable pre-training resources. 3) Given real images, apply an image editing model that takes specific types of instructions, such as color, style, and object change [4, 24].

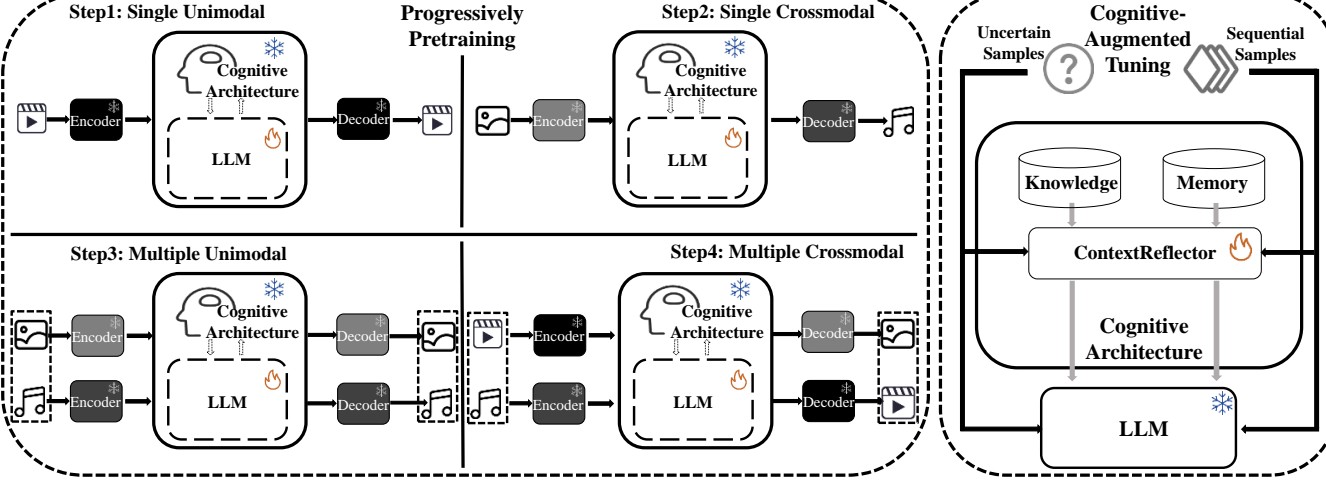

Figure 1: (Left) Progressively pretraining stage. (Right) Cognitive-augmented tuning stage.

## 3 WORLDGPT

WorldGPT is composed of three modules: multimodal encoders that uniformly represent various modalities, a Large Language Model (LLM) integrated with the cognitive architecture, and multimodal decoders that project the LLM's output into the desired modality space. After developing this MLLM structure 3.1, WorldGPT can naturally process multimodal states as well as utilize the inherent textual knowledge within LLM. Next, we adopt a progressively pretraining procedure 3.2 to learn any-to-any state transition patterns from WorldNet-Wild.

### 3.1 MLLM as Foundation of World Model

Pretrained LLMs [1, 40, 41] contain extensive world knowledge summarized by human, making itself generalist world models. However, its abilities are constrained to textual tasks. To enable LLM processing multimodal information, we adopt the state-of-the-art language-enteric multimodal encoder LanguageBind [57] as the bridge between language and other modalities. The details are presented as follows.

For multimodal encoding, we first utilize LanguageBind to obtain a unified representation for the multimodal state. Then, similar with other MLLMs [15, 46], a linear projection layer will be applied to convert features from multimodal embedding into LLM embedding. To help LLM better utilize multimodal information, we introduce special tokens to specify the multimodal inputs. For example, a state represented by both video and audio will be transformed into the following sequence "<VID> [video embedding] </VID> <AUD> [audio embedding] </AUD>" before sending to LLM.

For multimodal decoding, inspired by recent work [54], we include special tokens into LLM's vocabulary as multimodal signals. When these special tokens have been decoded exhaustively, i.e., "[<IMG1><IMG2>...<IMGn>]", we decode the corresponding hidden states with a trainable transformer-based projection layer. In

this paper, we investigate two types of target output space: original modality space and language-aligned feature space.

### 3.2 Progressively State Transition Training

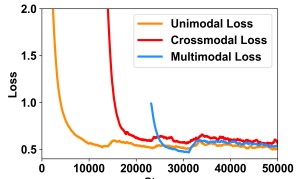 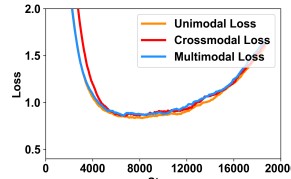

Figure 2: Loss for three types of tasks during progressively state transition training.

Figure 3: Loss for three types of tasks during naive state transition training.

WorldGPT is designed to solve state transition tasks with arbitrary modality as input and arbitrary modality as output. This requires a more delicate training paradigm, as training with mixture modality could cause unexpected loss escalation easily [28]. Therefore, we propose a *gentle* pretraining method, inspired by previous studies in Curriculum Learning [2]. Left part of Figure 1 briefly illustrates the training process.

Specifically, we train WorldGPT in an easy-to-hard paradigm by dividing the modality combinations into four types according to the learning difficulty: *single&unimodal*, *single&cross-modal*, *multiple&unimodal*, *multiple&cross-modal* and progressively introducing new combinations during training. As shown in Figure 2, by conducting such *gentle* training strategy, the learning curve is converged smoothly, and even the most challenging multimodal combination problems can eventually be handled just like unimodal ones after progressive training. We have also implemented the naive state transition training, where models are trained to predict *multiple&cross-modal* transition containing all combinations at first.

According to Figure 3, this results in a non-converging training process which highlights the necessity of progressively training.

We pick Vicuna-7B-v0 [40, 55] as the base LLM. Since the purpose of this stage is to build a generalist world model in this stage, therefore we only include WorldNet-Wild. And the whole cognitive architecture is not involved, only the LLM within WorldGPT is efficiently trained with only a small subset of parameters tuned utilizing LoRA [19] technique, which effectively decreases the number of trainable parameters. For each modality besides language, we choose LanguageBind which utilize a 24-layer, 1024-dimensional vision transformer with a patch size of 14 to encode multimodal states, and add 16 unique tokens into LLM's output vocabulary as modality signal. The whole training procedure is conducted on 8 × 80GB NVIDIA A100 GPUs, with a total batch size of 256. We train for 16 epochs, with each epochs trained on 1M samples. In the first 4 epochs, we only require single&unimodal predictions. More difficult modality compositions are gradually introduced every 4 epochs, with all compositions involved in the last 4 epochs.

More details about the training process can be found in the Supplementary Materials.

## 4 COGNITIVE ARCHITECTURE

After pretraining on WorldNet-Wild, WorldGPT is already a strong world model since it has seen various scenes and actions from millions of videos. However, in practical applications, there will inevitably be specific tasks that were not included in the pretraining dataset, resulting in sub-optimal performance. Therefore, we arm WorldGPT with the cognitive architecture 4.1, which enables WorldGPT to utilize external knowledge as well as past predictions through the ContextReflector. The ContextReflector is trained through Cognitive-Augmented Tuning 4.2 on WorldNet-Crafted with continuous prediction tasks and knowledge-augmented tasks.

### 4.1 Details of the Cognitive Architecture

**Working memory mechanism** will be activated when dealing with long-sequence prediction task. As shown in left part of Figure 4, past states, actions as well as predictions will be managed by the memory system. For future predictions, the memory system is capable of responding with historical information across any time horizon, including any content in any form (raw or reflected or other processed format), as long as it is requested by WorldGPT. Through utilizing history, WorldGPT will be able to generate temporal-consistent predictions more easily. For example, to simulate a first-person cooking case where the subject of the state constantly switches during the process, predicting next states solely on current state may lead to some mistakes such as displacement of objects. Utilizing historical infomation would help to maintain the consistency.

**Knowledge retrieval system** manages external knowledge, such as collections of state transitions, and supports retrieval actions through dense retrieval. All states and actions within these collections are encoded by a unified encoder, enabling efficient retrieval. When applied to a new domain, WorldGPT enhances its capabilities by retrieving the most similar experiences from the pre-collected knowledge base. For instance, consider a scenario where WorldGPT is employed as a simulator of a chemical laboratory, an environment replete with unique cases not encompassed in the training datasets, such as chemical reactions. If similar cases can be supplied by the knowledge retrieval system, WorldGPT could effectively predict outcomes by "imitating" the retrieved samples.

**ContextReflector** serves as an information extractor. Given an retrieved experience as context, the reflector will analyze the dynamics and derive the relevant knowledge based on the condition (i.e., current state and action). Specifically, as illustrated in the right part of Figure 4, the reflecting process is conducted in following steps: (1) Project the condition to the same dimension of learnable queries using a trainable linear layer. (2) Add the projected embedding with each query, obtaining a new set of conditional queries. (3) Send conditional queries and context embedding to the Querying Transformer [23], where the conditional queries will extract information from the context embedding by doing cross attention. (4) The extracted information (i.e. context tokens) will attach to the front of the inputs (i.e. current state and action) and send to WorldGPT. We built ContextReflector upon Q-Former [23] and train it from scratch. Considering ContextReflector is designed to extract task-relevant information rather than generic information, we only use 4 learnable queries which will produce 4 context tokens for each context.

Leveraging the cognitive architecture, past prediction histories as well as task-relevant knowledge can be automatically retrieved, further reflected to assist WorldGPT's prediction.

### 4.2 Cognitive-Augmented Tuning

As illustrated in the right part of Figure 1, we construct uncertain samples for knowledge-enhanced training and sequential samples for memory-enhanced training. Considering data quality and a more concentrated data distribution, all samples are derived from WorldNet-Crafted. Specifically, to generate memory-augmented samples, we first randomly determine the history length for each state within long sequences, ranging from 2 to 5. Then, only historical information of the specified length will be included in the training samples. For the creation of knowledge-augmented samples, we randomly select 200 samples from each scenario within the WorldNet-Crafted dataset to serve as the knowledge base, with the remain of the data served for training. Throughout the training process, each sample is augmented by retrieved knowledge from the corresponding scenario knowledge base. Our goal is to train WorldGPT to utilize the knowledge as well as memory through the ContextReflector, therefore we keep the entire LLM (including LoRA parameters) frozen and train ContextReflector only to extract useful information. The training is conducted on 8 × 80GB NVIDIA A100 GPUs, with a total batch size of 128. We totally train 2 epochs, with each epoch trained on 1M samples.

## 5 WORLDNET

To foster research in building realist world models, we manually collect state transition datasets from various source and construct a comprehensive dataset named **WorldNet**. WorldNet consists of two subsets: WorldNet-Wild and WorldNet-Crafted. WorldNet-Wild encompasses millions of samples from a broad spectrum of real-world scenarios and tasks, ensuring comprehensive coverage.

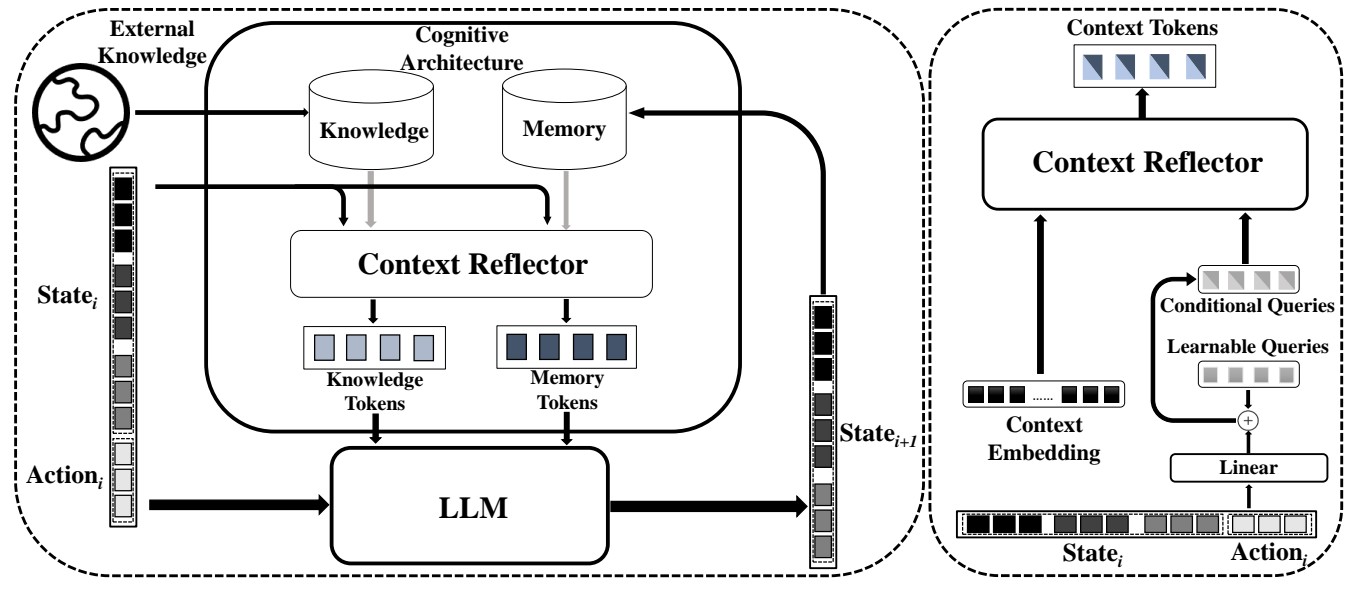

**Figure 4: (Left) The working flow of WorldGPT and cognitive architecture. (Right) The detailed model architecture of ContextReflector.**

Conversely, WorldNet-Crafted focuses on high-quality data, featuring specialized scenarios, extended sequence tasks, and meticulous annotations. The data statics of WorldNet is listed in Table 2. Some representative cases across six scenarios are plotted in Figure 5.

**Table 2: Detailed statistics of WorldNet.**

| | Videos | | | Num. Modal Compositions | | |
|---|---|---|---|---|---|---|
| | Num. | Avg. Modal | Avg. Words. | Single Cross | Multi Uni | Multi Cross |
| Wild | 11M | 2.61 | 7.15 | ≈60M | ≈40M | ≈400M |
| Crafted | 0.3M | 1.78 | 5.67 | ≈1.3M | ≈0.9M | ≈10M |

## 5.1 Constructing WorldNet-Wild from Unlabelled Videos

Existing video datasets with fine-grained action annotations are relatively small in scale and constrained to specific domains. To improve WorldGPT's generalization ability, we turn up to utilize unlabelled public videos sourced from internet, then leveraging dense video captioning method (i.e. Vid2Seq [48]) to generate the time intervals and specific event descriptions occurring in the video. The derived samples will be further filtered with the criteria being reasonable time length and reasonable action description complexity. All activity descriptions will be rewritten by ChatGPT [1] to ensure clean and correct in grammar. The whole procedure produces over ten million state transition samples, with most represented by all three modalities: video, audio and image (sampled from video). We name this dataset WorldNet-Wild.

WorldNet-Crafted is collected from following datasets: YT-Temporal-180M [53] and HowTo100M [30] and More details about these datasets can be found in the Supplementary Materials.

## 5.2 Constructing WorldNet-Crafted from Various Sources.

We manually collect human-labelled action datasets covering a wide range of domains, including cooking [56], egocentric activities [10], etc. These datasets are generally focused on some specific scenarios, together with detailed annotations of the action descriptions and time interval. What's more, some datasets are initially designed to pay more attention on a specific modality, such as audio in AVQA [50]. After aggregating all datasets, we briefly define six scenarios based on their original annotations, namely cooking, domestic work, entertainment, outdoor activity, sports and labor. Then we leverage ChatGPT to classify the most appropriate scenario for each sample within the collected datasets. We preserve the original temporal context of the sequential samples by retaining their annotations. This curated dataset is named as WorldNet-Crafted.

WorldNet-Crafted is collected from following datasets: Ego4D [10], Something-Something V2 [9], YouCook2 [56], AVQA[50] and Charades [37]. More details about these datasets can be found in the Supplementary Materials.

## 6 APPLYING WORLDGPT AS A MULTIMODAL INSTRUCTION SYNTHESIZER

To enhance the ability of multimodal agents to follow instructions, recent studies have developed various methods for synthesizing multimodal instructions [4, 24, 27, 29]. However, given that existing visual generators are only conditioned on descriptions (e.g., Stable Diffusion [33]) or simple editing instructions (e.g., InstructPix2Pix [4]), all these methods are limited to synthesizing instructions suitable for specific tasks, such as image classification or image editing. In this paper, we investigate WorldGPT as a universal world simulator, capable of interacting with agents through a variety of actions. The high degree of interactivity naturally expands the diversity of

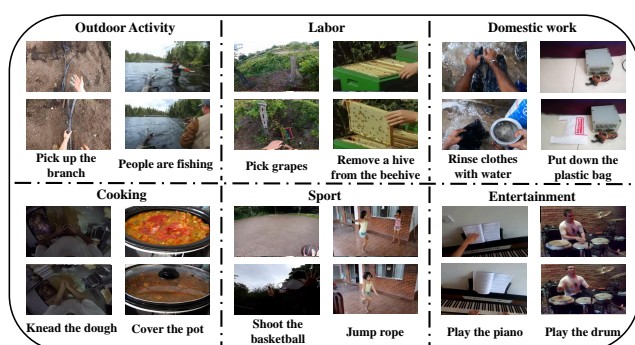

**Figure 5: Representative cases selected from WorldNet. WorldNet contains state transition samples across diverse domains.**

instructions. For example, as shown in Figure 6, WorldGPT combined with GPT-4 can generate complex, image-text interleaved recipes. By fine-tuning downstream agents on these synthetic instructions (we call it **dream tuning** since the whole scenario comes from WorldGPT's imaginations), task-specific knowledge can efficiently transferred from WorldGPT to agents. The generation pipeline consists of three steps: 1) textual instruction generation 6.1, 2) multimodal instruction completion 6.2, and 3) filtering and post-processing 6.3.

## 6.1 Textual Instruction Generation

Given a small seed set of instructions, similar with Self-Instruct [44], we first initialize the instruction pool with seed instructions. Then for every step, we randomly pick 8 instructions from pool as in-context examples. Unlike Self-Instruct, which restricts the proportion of original and synthesized instructions in the context, we do not impose this limitation hoping to enhance the diversity of the synthesized dataset, considering that multimodal instructions are inherently richer than pure text instructions. Then We use GPT-4 [1] to generate novel instructions based on in-context examples.

## 6.2 Multimodal Instruction Completion

After obtaining the textual part of the instruction, we further utilize WorldGPT to complete the multimodal part. The current version of WorldGPT supports two types of input as condition: 1) textual state description, similar with text to image/video/audio model and 2) past state and current action. Therefore, before sending to WorldGPT, we use GPT-4 for another time to identify the instruction pattern and extract states and actions from instructions. For example, as shown in Figure 6, when it's required to synthesis recipes similar with samples from YouCook2, the first step of the recipe will be considered as the description of the first state while the rest of the steps will be treated as actions which trigger transition to new states. After extracting entities from textual instruction as conditions, we use WorldGPT to generate the corresponding content. For long-sequence prediction tasks, past prediction histories will be utilized through ContextReflector to maintain temporal consistency. Considering most multimodal agents take the multimodal inputs using a language-aligned multimodal encoder (e.g.,

CLIP [32]), the output of transformer-based projection layer will map the signal back to WorldGPT's unified multimodal encoding space (i.e. LanguageBind feature). During dream tuning, it would be highly efficient by using a lightweight projector (e.g. linear layer) to connect the output of WorldGPT to agent's input encoding space which saves both rendering and encoding time.

## 6.3 Filtering and Post-processing

To ensure the diversity of the curated instruction pool, we preemptively calculate the similarity between new instructions and existing ones, aiming to filter out repetitive content. In early experiments, we observed that instructions similar in text could have significant differences in their multimodal components, thus maintaining good diversity. Therefore, our criterion for determining the similarity of instructions from a textual perspective is whether the ROUGE-L overlap exceeds 0.8, which is relatively lenient compared with [44]. In early experiments, we also tried using methods like CLIP similarity to calculate the similarity of the multimodal components. However, due to the high computational complexity and the lack of significant improvement in the quality of the selected instructions, we ultimately considered only the textual part of the instructions during filtering.

## 7 EVALUATION

To comprehensively evaluate WorldGPT's potential as a generalist world model, we conducted the following two experiments: 1) State transition ability benchmarking 7.1 on WorldNet-Crafted, which encompasses evaluations of cross-modal, modal combination, long-sequence, and knowledge-enhanced capabilities. 2) Quality assessment of synthesized instructions 7.2 by comparing the performance of fine-tuned multimodal agents on three categories of tasks: visual understanding, embodied planning, and audio-video question answering.

## 7.1 Benchmarking State Transition on WorldNet-Crafted

**Experiment Setting.** Before training, we pre-sample a subset from the dataset to serve as the test set, ensuring that the test set never appeared during the training phase, neither as samples nor as part of the knowledge base. For each scenario within the six scenarios, we collect 25 samples as the test dataset and re-select 50 samples from the training dataset as the knowledge base. For the continuous prediction task, we select a total of 200 sequential samples originally collected from Ego-4D and YouCook2 . All sequences are uniformly set to a length of 7. As for the evaluation metric, we use the cosine similarity between the predicted results of each model and the real data in the model's encoding space.

**Baseline.** For WorldGPT, we consider itself (without cognitive architecture) and its four variants:

- **WorldGPT + CK**, which stands for "in context knowledge," meaning the retrieved knowledge embedding will be directly appended before the input state embedding.
- **WorldGPT + RK**, which stands for "reflected knowledge," meaning the retrieved knowledge embedding will be first sent to ContextReflector, then appended.

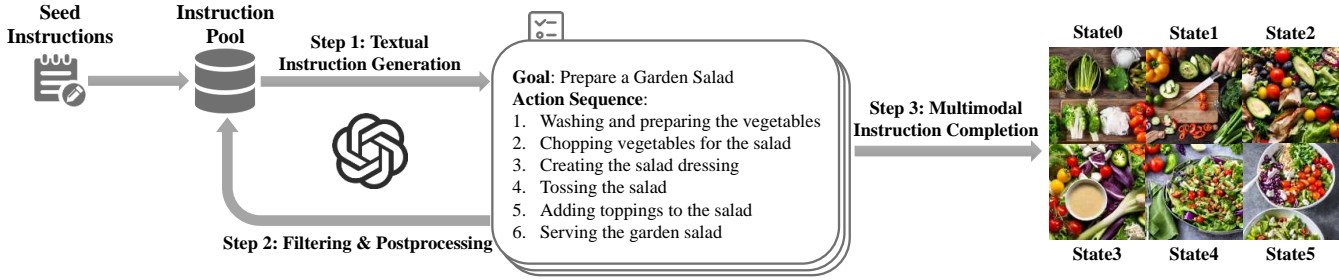

**Figure 6: The process of constructing a multimodal instruction pool.**

**Table 3: Evaluation results for different modality combination inputs and outputs, with the best values highlighted in bold.**

| | Unimodal | | | Cross-modal | | | | | | All | | |
|---|---|---|---|---|---|---|---|---|---|---|---|---|
| *Input Modality* | *image* | *video* | *audio* | *image&audio* | *video&audio* | *image* | | *video* | | *image&video&audio* | | |
| *Output Modality* | *image* | *video* | *audio* | *video* | *image* | *video* | *audio* | *image* | *audio* | *image* | *video* | *audio* |
| CoDi [39] | 62.6 | 65.8 | 21.3 | 52.4 | 57.6 | 58.3 | 13.0 | 54.9 | 10.2 | 62.7 | 62.6 | 16.7 |
| NexT-GPT [46] | 57.1 | 62.4 | 26.5 | 41.5 | 53.6 | 49.1 | 22.5 | 56.9 | 28.4 | 53.5 | 59.6 | 28.1 |
| WorldGPT | 71.6 | 72.2 | 45.6 | 58.0 | 79.2 | 65.2 | 41.7 | 79.6 | 34.6 | 78.0 | **82.7** | 37.1 |
| WolrdGPT+CK | 72.4 | 72.0 | 44.3 | 58.3 | 79.1 | 65.6 | 41.2 | 76.1 | 33.4 | 75.7 | 74.1 | 34.1 |
| WorldGPT+ RK | **75.6** | **76.4** | **50.1** | **62.7** | **81.5** | **71.6** | **45.3** | **82.4** | **43.6** | **80.1** | 82.5 | **42.4** |

- **WorldGPT + CM**, which stands for "in context memory," meaning the retrieved memory embedding will be directly appended before the input state embedding.
- **WorldGPT + RM**, which stands for "reflected memory," meaning the retrieved memory embedding will be first sent to ContextReflector, then appended.

For comparison, we select two multimodal models that also support any-to-any generations:

- **CoDi** [39], a diffusion-based generative model capable of generating any combination of output modalities, such as text, image, video, or audio, from any combination of input modalities.
- **NExT-GPT** [46], an end-to-end general-purpose any-to-any multimodal Large Language Model. By connecting an LLM with multimodal adaptors and different diffusion decoders, NExT-GPT is capable of understanding inputs and generating outputs in any combination of language, images, videos, and audio.

We hope to conduct similar evaluations on other advanced realist world models, such as UniSim and WorldDreamer. Unfortunately, before the submission deadline, none of these models were open-sourced or replicable. We plan to extend this evaluation in the future.

**Evaluating Result of Different Modality Compositions.** As shown in Table 3, we evaluated eight modality combinations, comprising three basic unimodal prediction tasks, four cross-modal prediction tasks, and one all-to-all prediction task. Based on the results, we delineate three key insights:

- WorldGPT consistently outperforms CoDi and NeXT-GPT by a significant margin across all tasks, which underscores its superior capability in modeling complex interactions within the world.

- WorldGPT effectively handles multimodal tasks, showing strong performance in modal combination input, cross-modal generation, and joint generation tasks.
- The prediction accuracy of WorldGPT for audio tasks is significantly lower than that for video and audio tasks. This may be due to the data collection being primarily visually oriented, resulting in weaker causality in the audio modality.

**Table 4: Evaluating performance on long sequence samples using cosine similarity**

| Sequence Length | 1 | 3 | 5 | 7 |
|---|---|---|---|---|
| WorldGPT | 72.5 | 72.3 | 72.6 | 73.1 |
| WorldGPT+CM | 72.5 | 72.1 | 71.8 | 69.6 |
| WorldGPT+RM | **72.5** | **74.1** | **74.4** | **73.8** |

**Ablation Study of ContextReflector.** For the knowledge-augmented prediction task, it is evident from Table 3 that WorldGPT+RK achieves the highest scores in the majority (11 out of 12) of cases. Similar outcomes are observed in Table 4, where WorldGPT+RM consistently achieves the highest similarity scores in all cases. We attribute this success to ContextReflector, which effectively extracts useful knowledge from retrieved samples and temporal information from previous history. However, we also note that the performance gain may diminish when processing excessively long input sequences, such as those combining three modalities with reflected embeddings, or for the 7th samples in sequential predicting. This suggests a potential overload for WorldGPT's processing capacity.

## 7.2 Evaluating WorldGPT as a World Simulator

**Constructing Tasks based on WorldNet-Crafted.** In order to comprehensively evaluate the quality of the synthesized instructions, we manually design three tasks based on WorldNet-Crafted:

- *Visual Understanding* requires models to output the changes between two visual states. All samples are selected from Something-Something V2 [9] and Charades [37].

**Table 5: Evaluation results of baseline models across three tasks. The best result is in bold, and the second best is underlined. DM is short for Diffusion Model.**

|  | Zero-shot | Fine-tuned | | |
|---|---|---|---|---|
|  |  | Real Data | WorldGPT | DM |
| *Visual Understanding* | | | | |
| MiniGPT-4 [58] | 15.0 | **36.3** | 34.8 | 23.8 |
| mPLUG-Owl [52] | 13.0 | **32.3** | 31.5 | 20.0 |
| *Embodied Planning* | | | | |
| Video-LLaVA [25] | 12.0 | **40.5** | 38.3 | 18.8 |
| mPLUG-Owl [52] | 15.8 | 37.8 | **38.8** | 21.5 |
| *AVQA* | | | | |
| ImageBind-LLM [16] | 18.0 | 47.5 | **48.5** | 21.0 |

- *Embodied Planning* requires models to plan future actions based on historical egocentric video records. All samples are selected from Ego-4D [10] and YouCook2 [56]. Similar to EgoPlan-Bench [7], we utilize ChatGPT [1] to process original annotations to obtain goal-oriented action sequences.
- *Audio-Video Question Answering* requires the model to answer questions based on information organized in both audio and video. Here, we directly use the annotations from the original AVQA [50] dataset.

Then, from the transformed set of task instructions, we selected 4,000 as the authentic training set, 400 as the test set, and an additional 200 as the seed instruction set, which will be used to synthesize 4,000 instructions with WorldGPT.

**MLLM Agents**. In this experiment, we employ four MLLM agents: MiniGPT-4 [58], mPLUG-Owl [52], Video-LLaVA [25], and ImageBind-LLM [15]. For the *Visual Understanding* task, we test MiniGPT-4 and mPLUG-Owl, both of which are commonly used baselines in research. For the *Embodied Planning* task, we test Video-LLaVA and mPLUG-Owl, both of which support processing and training on video inputs. For the *Audio-Video Question Answering* task, we test ImageBind-LLM, which encodes multiple modalities with a uniform ImageBind encoder. During the training and inference phases, we use the default configurations of these models.

**Baseline Setting.** We conduct experiments on following settings:

- Zero-shot: Direct evaluation using the original MLLM agents without any fine-tuning.
- Real data: Using authentic data to fine-tune MLLM agents.
- WorldGPT: Using synthetic instructions from WorldGPT to fine-tune MLLM agents.
- Diffusion Model: Using synthetic instructions from modality expert diffusion model to fine-tune MLLM agents. For image generation, we use Stable Diffusion [33] and InstructPix2Pix [4]. For video generation, we use Zeroscope[1] and StableVideo [6]. For audio generation, we use AudioLDM [26].

**Evaluation details.** We utilize ChatGPT to assess the correctness of the models' output. The design of the prompt is referenced from [35] and we plot it in the Supplementary Materials. Both the training and inference procedures are conducted on a single NVIDIA A100 80G GPU.

---

[1] https://huggingface.co/cerspense/zeroscope_v2_576w

**Table 6: The detailed generation time for WorldGPT and modality expert to complete multimodal instructions. DM is short for Diffusion Model.**

| Time (minute) | WorldGPT | DM |
|---|---|---|
| *Visual Understanding* | 50 | 134 |
| *Embodied Planning* | 47 | 2176 |
| *AVQA* | 80 | 2,449 |

**Results Analysis.** As shown in Table 5, agents trained on synthetic instructions from WorldGPT perform as effectively as those trained on authentic instructions, demonstrating the reliability of WorldGPT as a universal world model. Furthermore, agents trained on instructions generated by modality expert diffusion models show relatively small improvements over the original agents, indicating that these models lack an understanding of the world's dynamics. Additionally, WorldGPT has an absolute advantage in terms of efficiency, as illustrated in Table 6. For *Audio-Video Question Answering* tasks, WorldGPT is 30 times faster than traditional diffusion-based methods. We attribute this to the prediction of state transitions in the feature space, which avoids the time-consuming rendering process in the modality space. We plot some cases in Figure 7. The compared generation model failed to understand actions, whereas ours demonstrates powerful capabilities in learning world dynamics.

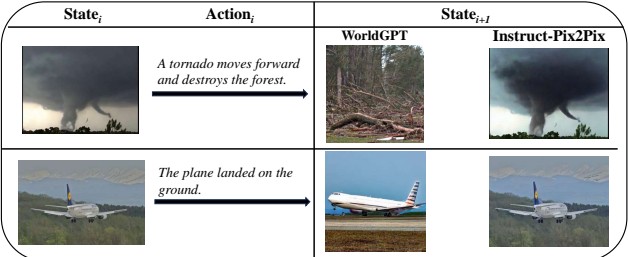

**Figure 7: Some representative state transition prediction cases.**

## 8 CONCLUSION

In conclusion, this paper introduces WorldGPT, a novel generalist world model which can understand and predict state transition across any combination of modalities. To improve WorldGPT's capability in specialized domain as well as long sequence prediction, we design a cognitive architecture which can automatically extract relevant information from external knowledge and history predictions. We construct a comprehensive multimodal transition dataset, namely WorldGPT, which can serve as both pretraining resources or evaluation benchmark. We further explore WorldGPT as a universal world simulator, which can transfer internal knowledge about the world dynamics to downstream agents through synthesizing instruction data.

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
