# OpenReview forum: "WorldGPT: Empowering LLM as Multimodal World Model"
_acmmm.org/ACMMM/2024/Conference — MM2024 Oral_

### Official Review · Reviewer_Luis · 2024-05-11

**Rating:** 4
**Confidence:** 1

**Summary:**

This paper introduces a generalist world model, WorldGPT, which predicts the next state by learning from millions of videos. Additionally, the authors propose a cognitive architecture to enhance the capabilities of WorldGPT. Furthermore, they construct a benchmark, WorldNet, to evaluate the model's performance and advance the field's future development.

**Strengths:**

The authors made a holistic effort towards a multimodal world model, including the design of a novel cognitive architecture, training of WorldGPT, and providing the benchmark WorldNet. Learning real-world knowledge from millions of videos also sounds interesting.

Since I have a limited background in the world model, I will determine my final opinion according to the feedback from other reviewers.

**Limitations:**

I have several questions about the settings in this paper.

Q1. What's the relationship between the WorldGPT and the "world model" introduced by LeCun? In my understanding, WorldGPT is able to take multimodal inputs and output multimodal results. This concept sounds similar to the NExT-GPT, but they do not claim to be the world model.

Q2. Continuing from the above question, what's the main difference between WorldGPT and NExT-GPT? Additionally, the NExT-GPT evaluates their model in various tasks, e.g. text-to-image, can WorldGPT be evaluated in this setting?

Q3. In section 7.1, WorldNet uses the cosine similarity between prediction and groundturth in the model’s encoding space. Is this a standard practice for evaluation? Why not directly compare the difference between predicted and true values?

**Suitability:**

3

---

### Official Review · Reviewer_nPDT · 2024-05-24

**Rating:** 6
**Confidence:** 4

**Summary:**

This work proposes a generalist model (WorldGPT) based on multimodal LLM. The WorldGPT is enhanced in specialized scenarios with a novel cognitive architecture. In addition, the author also designs a WorldNet for evaluation. The WorldGPT has a potential to be a world simulator.

**Strengths:**

Good organization and good writing.
This paper designs a generalist word model, which supports input and output with multiple modalities.
This paper proposes a WorldNet for world state transitions, ideal for training and evaluating world models.
This paper proposes a new learning paradigm to help agents acquire knowledge efficiently from WorldGPT.
The experimental results is sufficient.

**Limitations:**

The author can provide more comparisons with existing LLMs, for example, GPT4 and Lama to further demonstrate the efficiency.
The author can provide more visual comparisons for different modalities (maybe in the supplement materials).

**Suitability:**

3

---

### Official Review · Reviewer_VxuT · 2024-05-26

**Rating:** 3
**Confidence:** 3

**Summary:**

The paper introduces WorldGPT, a flexible model trained on millions of videos to understand and work with different types of data. It presents a new architecture for these models, which includes memory management and knowledge retrieval. The paper also creates WorldNet, a large dataset to help train and test these models. Additionally, it proposes a new learning method where agents can learn efficiently from WorldGPT using synthetic data.

**Strengths:**

The concept of leveraging state transitions is intriguing. The experiments evaluating the approach as a world simulator demonstrated the effectiveness of the methods proposed in the paper.

**Limitations:**

The paper would benefit from a discussion and comparison with recent works on state transitions. For instance:

Xue, Z., Ashutosh, K., & Grauman, K. (CVPR 2024). Learning Object State Changes in Videos: An Open-World Perspective.

Niu, Y., Guo, W., Chen, L., Lin, X., & Chang, S. (ICLR 2024). SCHEMA: State CHangEs MAtter for Procedure Planning in Instructional Videos.


Nguyen, N., Bi, J., Vosoughi, A., Tian, Y., Fazli, P., & Xu, C. (NAACL 2024). OSCaR: Object State Captioning and State Change Representation.


Additionally, it would be advantageous to conduct comparisons with more models besides CoDi and NextGPT.

Could you please clarify why, in Table 3 under the "All" setting, WorldGPT+CK performs significantly lower than both WorldGPT and WorldGPT+RK?

**Suitability:**

2

---

### Official Review · Reviewer_xqYs · 2024-05-27

**Rating:** 4
**Confidence:** 2

**Summary:**

This paper introduces WorldGPT, a sophisticated world model trained on unlabelled videos to predict state transitions and support multiple modalities. Additionally, the authors present WorldNet, a comprehensive dataset specifically designed for training and evaluating this world model. The proposed progressive learning method aids in model convergence and performance.

**Strengths:**

1. The paper is generally well-organized and clearly written, making it easy to follow the authors' arguments and findings.

2. The motivation is clear and well-founded.

3. The construction of the WorldNet dataset may make some contributions that could aid in furthering research and development within the community.

**Limitations:**

1. The experimental section could benefit from additional comparisons. For instance, including evaluations against close-resource Large Language Models for unimodal tasks would provide a more comprehensive assessment of WorldGPT’s performance.

2. The consistency of ID information is an issue, as evidenced by discrepancies like those in the second line of Figure 7, the plane before and after state transition is obviously different. The paper would benefit from a discussion on how to maintain or improve ID consistency.

3. The details of the Progressive State Transition training methodology require further elaboration. Specifically, the section would benefit from a more comprehensive breakdown of the training process, including the number of steps allocated to each task phase. Moreover, the explanations surrounding Figure 2 and Figure 3 are somewhat lacking. The loss fluctuation observed around the 300K step needs to be explained. Also, why the naive state transition training mode cannot converge needs to be discussed. Another aspect worth exploring is the potential for training optimization based on the difficulty of the input modalities,  e.g., only image-input, combination covering image and video input, and then all the modalities, which training method can achieve better performance.

**Suitability:**

3

---

### Meta-Review · Area_Chair_ooko · 2024-07-01

**Recommendation:** Accept (Oral)
**Confidence:** 5

**Metareview:**

The only reviewer who had judged on the reject side changed his decision to Borderline Accept, and all reviewers judged on the accept side.
It is appropriate to accept this paper.